# Neoadjuvant modified FOLFIRINOX plus nivolumab in borderline-resectable pancreatic ductal adenocarcinoma: a pilot phase 1 trial

Chemotherapy and immune checkpoint inhibitor combinations have failed to improve survival in pancreatic ductal adenocarcinoma (PDAC), except in rare microsatellite instability-high cases; most studies focused on advanced disease. Here, we present clinical and translational results from a single-arm, prospective phase 1b/2 investigator-initiated study (NCT03970252) evaluating neoadjuvant modified FOLFIRINOX (mFFX) plus nivolumab in patients with borderline-resectable PDAC. The co-primary endpoints of safety and pathological response rate were met, with 22 (79%) of 28 patients proceeding to surgery and no grade ≥3 immune-related adverse events. All grade 3-4 treatment-related adverse events were chemotherapy-related. By CAP scoring, 9% of patients achieved a complete pathologic response, 9% a near-complete response, and 72% a partial response. Secondary endpoints included CA 19-9 response rate, R0 resection rate, objective response rate, and disease-free survival (median 19.7 months, 95% CI: 7.3-30.8). In post-hoc analyses, median progression-free survival was 26 months (95% CI: 14.7-34.3), and median overall survival was 38 months (95% CI: 27.9-not reached). Exploratory gene expression, immunohistochemistry and spatial transcriptomics showed increased intratumoral plasma cells and CD8 T cells in treated patients versus mFFX-only controls, and lymphoid aggregates with high plasma-cell-to-B cell ratios enriched for terminally exhausted CD8 T cells with fewer progenitor exhausted CD8 T cells and central memory CD4 T cells.

Pancreatic ductal adenocarcinoma (PDAC) is one of the most aggressive malignancies, characterized by poor prognosis and limited therapeutic options[1]. Recent advancements in systemic chemotherapy, most notably FOLFIRINOX (leucovorin, fluorouracil, irinotecan, and oxaliplatin) have significantly improved survival in both early and advanced PDAC[2-4]. In contrast, immune checkpoint inhibitors (ICI) have shown minimal benefit in PDAC, except for the 1-2% of patients with microsatellite instability-high (MSI-H) tumors[5-7]. Even in MSI-H

PDAC, response rates are markedly lower than those seen in other MSI-H cancers[8].

In several other solid tumors, adding ICIs to neoadjuvant chemotherapy has led to meaningful improvements in clinical outcomes[9]. In the Phase 2 randomized NADIM-II trial for non-small cell lung cancer (NSCLC), the addition of nivolumab increased the pathologic complete response (pCR) rate from 7% to 37%, and extended 24-month progression free survival (PFS) from 41% to 67%[10]. Similarly, in the

✉e-mail: zwainberg@mednet.ucla.edu; jmlink@mednet.ucla.edu; tdonahue@mednet.ucla.edu

CheckMate 816 trial for NSCLC, the pCR rate rose from 2% to 24% with the addition of nivolumab to neoadjuvant chemotherapy[11]. In the I-SPY2 trial for high-risk breast cancer subtypes, pembrolizumab increased the pCR from 19% to 44%[12].

Borderline-resectable pancreatic cancer (BRPC) is characterized by tumor involvement of adjacent major blood vessels making resection technically feasible but more complex, with higher risk of recurrence than earlier-stage resectable disease[13]. The current standard-of-care for BRPC includes 3-6 months of neoadjuvant therapy, most commonly modified FOLFIRINOX (mFFX), to address both local and occult systemic disease. In contrast to patients with locally advanced disease and more extensive vessel involvement, the majority of BRPC patients proceed to surgery following preoperative therapy[13]. This clinical setting is particularly well-suited for evaluating novel therapeutics, as it enables assessment of treatment response and allows for comprehensive analysis of surgical resection specimens. These specimens provide a unique opportunity to study treatment-induced changes, mechanisms of action, and identify pathways of resistance.

Here, we report the safety, efficacy, and translational findings of a single-arm Phase 1b/2 trial evaluating mFFX with nivolumab for patients with BRPC. Recent trials in patients with BRPC combining PD-1 blockade with fluorouracil- or gemcitabine-based chemoradiation have demonstrated only limited clinical benefit[14,15]; however, none have evaluated PD-1 inhibition in combination with mFFX—the most effective chemotherapy regimen in PDAC. To evaluate the immunologic and pathologic effects of this combination, we compare post-treatment resection specimens to patient-matched pretreatment biopsies and analyzed stage-matched historical controls treated with mFFX alone to isolate the contribution of PD-1 blockade. Through integration of clinical outcomes, molecular profiling, immunohistochemistry, and single-cell spatial transcriptomics, we identify undesirable treatment-associated immune cell changes and abnormalities in intra-tumoral lymphoid aggregates (LA) that may be targeted in future therapeutic strategies.

## Results

### Patient characteristics

Between August 2019 and March 2023, 35 patients with biopsy-proven borderline-resectable pancreatic adenocarcinoma (BRPC) were screened for eligibility. Of these, 28 patients (80%) were enrolled and received neoadjuvant mFFX and nivolumab (anti-PD-1) as shown in Fig. 1. All 28 patients who received ≥1 dose of study treatment were included in the analyses. Six patients did not proceed to surgery, 4 due to disease progression. All patients who underwent surgery had their tumors resected. Detailed demographics and clinical characteristics of all patients are provided in Table 1. The median age was 67.5 years, and all patients had an ECOG performance status of 0 or 1. Patients received a median 5.5 cycles of treatment (range 1-6).

### Safety and Tolerability

As per the protocol, interim safety assessments were performed when the first 7 and 15 evaluable patients were enrolled in the study. This revealed no dose limiting toxicities and the summary of treatment-related adverse events (TRAEs) is given in Table 2. At least one grade 3 or higher AE deemed related to treatment occurred in 11% of patients. There were 4 grade 4 AEs (3 neutropenia, and one hypokalemia). There were no ≥grade 3 AEs related to nivolumab and no ≥grade 3 immune related adverse events (irAEs). All surgeries were performed without treatment-related delays, per protocol. There were no Grade C pancreatic fistulas, meeting the pre-specified primary safety endpoint of the study.

### Efficacy and Surgical Outcomes

Treatment efficacy was evaluated 3 months after the first cycle using protocol-defined secondary endpoints, including RECIST v1.1 criteria

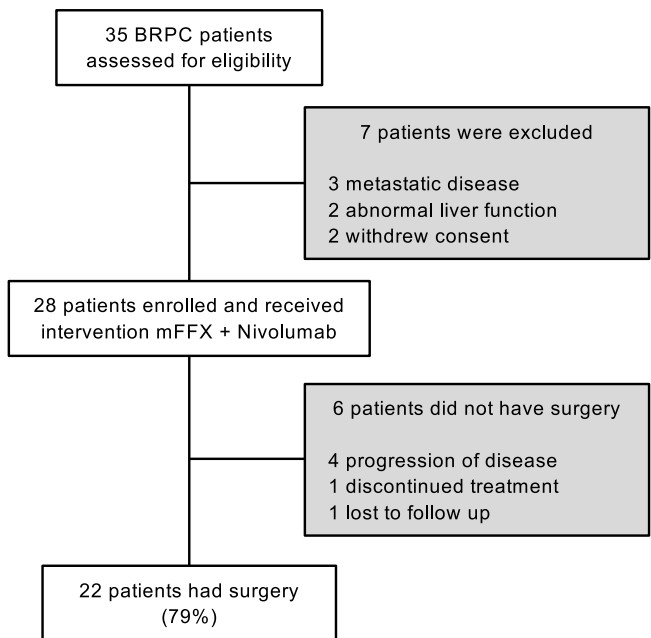

**Fig. 1 | CONSORT flow diagram of patient eligibility, enrollment, and surgery.** CONSORT diagram of all patients assessed for eligibility, treatment, and progression to surgery.

## Table 1 | Patient Demographics and Baseline Characteristics

| Characteristic | N = 28 |
| --- | --- |
| Median age, years (range) | 67.5 (44,79) |
| Sex, n (%) | |
| Male | 16 (57) |
| Female | 12 (43) |
| Race and ethnicity, n (%) | |
| White | 16 (57) |
| Asian | 6 (21) |
| Middle Eastern or North African | 1 (4) |
| Hispanic or Latino | 4 (14) |
| American Indian/Alaska Native | 1 (4) |
| Baseline ECOG PS score, n (%) | |
| 0 | 17 (61) |
| 1 | 11 (39) |
| Median number cycles completed (range) | 5.5 (1,6) |
| Best response, n (%) | |
| Stable disease | 19 (68) |
| Partial response | 7 (25) |
| No imaging performed | 2 (7) |

*ECOG PS* Eastern Cooperative Oncology Group Performance Status.

of the single target pancreatic lesion and serial CA19-9 levels measured at baseline and every 4 weeks thereafter. CA19-9 decreased from the first to the last cycle for 19 of 26 (73%) patients completing treatment, including 8 (31%) patients who converted from abnormal (> 37 units/ml) to normal levels (Fig. 2A). Imaging-based primary tumor size decreased for 21 (81%) patients, with 7 (27%) patients achieving a partial response (the overall response rate), and 19 (73%) patients with stable disease (SD) as their best response (Fig. 2B). The surgical resection rate of 79% (22 of 28) among enrolled patients exceeded the 58% reported in the randomized ALLIANCE A021501 trial[16].

The timing of surgery was determined through multi-disciplinary review. At the time of surgery, 69% of tumors were centered in the

## Table 2 | Safety and Tolerability

| Hematologic toxicities, N = 28 | Grade 1, n (%) | Grade 2, n (%) | Grade 3, n (%) | Grade 4, n (%) |
|---|---|---|---|---|
| Anemia | 0 (0) | 3 (11) | 2 (7) | 0 (0) |
| Hypokalemia | 2 (7) | 5 (18) | 3 (11) | 1 (4) |
| Hypomagnesemia | 1 (4) | 1 (4) | 1 (4) | 0 (0) |
| Hyponatremia | 0 (0) | 0 (0) | 1 (4) | 0 (0) |
| Neutrophil count decreased | 0 (0) | 0 (0) | 2 (7) | 3 (11) |
| Platelet count decreased | 6 (21) | 3 (11) | 1 (4) | 0 (0) |
| Nonhematologic toxicities, N = 28 | Grade 1, n (%) | Grade 2, n (%) | Grade 3, n (%) | Grade 4, n (%) |
| Dehydration | 1 (4) | 10 (36) | 0 (0) | 0 (0) |
| Diarrhea | 18 (64) | 7 (25) | 0 (0) | 0 (0) |
| Musculoskeletal and connective tissue disorder – muscle aches | 6 (21) | 0 (0) | 1 (4) | 0 (0) |
| Nausea | 24 (86) | 9 (32) | 0 (0) | 0 (0) |
| Neuropathy | 17 (61) | 1 (4) | 0 (0) | 0 (0) |
| Vomiting | 11 (39) | 5 (18) | 0 (0) | 0 (0) |
| Watering eyes | 1 (4) | 0 (0) | 0 (0) | 0 (0) |
| Weight loss | 1 (4) | 2 (7) | 0 (0) | 0 (0) |

pancreatic head or uncinate and 23% in the distal pancreas (Table 3). A limited vascular resection (venous and/or arterial) was performed in 45% of cases ($n = 10$). Postoperative complications included only two ISGPS Grade A/B fistulas; no Grade C fistulas occurred.

On histopathologic analysis, 50% of patients had node-negative disease (pN0) and 86% had no tumor at surgical margins (a pre-specified secondary endpoint of the trial). Using the College of American Pathology (CAP) treatment response scoring system, 2 patients (9%) achieved a complete pathologic response (CAP 0), 2 (9%) had a near complete response (CAP 1), and 16 patients (72%) had a partial response (CAP 2), meeting a pre-specified co-primary endpoint of the trial. One patient had locally advanced disease at the time of surgery.

A swimmer plot depicting treatment timelines and outcomes for each of the 26 patients is shown in Fig. 2C. Median disease-free survival was 19.7 months (95% CI: 7.3-30.8). The median disease-free survival from resection (a pre-specified secondary endpoint) was 19.7 months (95% CI: 7.3-30.8) (Fig. 2D). Median progression-free survival was 22 months (95% CI: 14.5-29.4) for all patients and 26 months (95% CI: 14.7-34.3) for patients who underwent resection (Fig. 2E). Median overall survival (OS) was 35 months (95% CI: 22.0-not reached) for all patients and 38 months (95% CI: 27.9-not reached) for those who underwent surgical resection (Fig. 2F). Median follow-up for surviving patients was 48 months. Individual patient outcomes are detailed in Supplementary Data. Germline genetic alterations for all patients (where available) are reported in the Supplementary Data.

### CD8 T cells and plasma cells are both more abundant after mFFX and nivolumab treatment than after mFFX alone

To assess leukocyte composition, RNA-sequencing (RNA-seq) was performed on 6 pre-treatment biopsies and 21 post-treatment resection specimens (including 5 matched pairs) from trial patients, as well as 9 resection specimens from stage-matched patients with BRPC who were treated with neoadjuvant mFFX alone.

Tumor infiltrating leukocyte populations were quantified using CIBERSORT-ABS deconvolution of the RNA-seq data. Post-treatment tumors harbored significantly more leukocytes ($P = 0.028$) and more lymphocytes ($P = 0.065$) compared to pre-treatment biopsies from the trial (Fig. 3A). However, the overall abundance of leukocytes and lymphocytes were similar between post-treatment trial tumors and those treated with mFFX alone. CD8 T cells were more abundant in post-

treatment trial tumors than those from patients who received mFFX alone ($P = 0.058$, Fig. 3B); though higher CD8 T cell scores were only modestly associated with improved patient outcomes ($P = 0.142$, Fig. 3C).

In melanoma patients treated with anti–PD-1 therapy, increases in intra-tumoral CD8 T cell density predict superior clinical responses[17], yet intra-tumoral CD8 T cell density alone is not associated with better outcomes in PDAC[18]. Thus, we considered that the nivolumab-related increase in CD8 T cells in our trial may be accompanied by dysfunction in other immune compartments. Among all leukocyte subsets, CIBERSORT-ABS revealed significant nivolumab-related differences in only 2 cell types: activated memory CD4 T cells (lower with nivolumab) (Fig. 3D) and plasma cells (PCs) (higher with nivolumab) (Fig. 3E). Tumors with CD8 T cell scores >0 were significantly more likely to have PC scores greater than the mean ($P = 0.03$, Fishers exact test) (Fig. 3F), suggesting concomitant increase of CD8 T cells and PCs by nivolumab.

### The plasma cell phenotype is more common in classical-subtype tumors from FFX+nivolumab treated patients

We applied PurIST[19] to RNA-seq data to assign classical and basal-like transcriptional subtypes of PDAC. Consistent with prior studies[20–22], patients with basal-like tumors fared worse than those with classical tumors (Supplementary Fig. 1A). Two of the 5 tumors classified as classical pre-treatment were assigned to the basal-like subtype after treatment (Supplementary Fig. 1B), indicating molecular progression during therapy.

Consistent with a recent report[23], plasma cell (PC) scores were higher in the classical-subtype tumors ($P = 0.055$) following treatment with mFFX and nivolumab (Supplementary Fig. 1C). Using DeCAF, we found that the tumor-restrictive CAF phenotype, associated with better prognosis (Supplementary Fig. 1D), was more common in classical-subtype than basal-subtype tumors (Supplementary Fig. 1E), and had numerically higher PC scores than the tumor permissive CAF subtype ($P = 0.050$, Supplementary Fig. 1F).

### Intra-tumoral PC density correlates with intra-tumoral lymphoid aggregate (LA) density after mFFX and nivolumab

Responses to ICIs have been associated with the presence of lymphoid aggregates (LAs), including mature tertiary lymph structures (TLSs) that are thought to prime immune responses against tumor-associated antigens and can drive B cell differentiation to PCs[24–26]. Given the observed increase in PCs after mFFX + nivolumab, we performed IHC on 20 resected tumors from trial patients to investigate whether specific LA characteristics might explain this increase.

IHC-quantified intra-tumoral, MUM1 + PC density correlated with CIBERSORT-ABS PC scores (Fig. 4A). Compared with mFFX-only tumors, those treated with mFFX + nivolumab had significantly higher intra-tumoral PC density (Fig. 4B), consistent with the bulk RNA-seq findings (Fig. 3E).

LAs were identified by CD20 + B cell clusters with adjacent CD3 + T cells (Fig. 4C). A total of 446 LAs were identified (median, 19.5 per tumor; average area of 0.07 mm2 ± 0.09 mm²; mean of 746 ± 935 cells). LAs were classified as either intra-tumoral or extra-tumoral based on histologic assessment of the treatment bed of the tumor (Supplementary Fig. 2A, B). LA area, total cell number, and CD20+ cell number were similar between locations (Supplementary Fig. 2C). However, intra-tumoral LAs had significantly lower CD20+ cell density than extra-tumoral LAs ($P < 0.001$) (Fig. 4D). Because high B cell density is associated with germinal center activity, we further assessed each LA for CD23 expression, a marker of follicular dendritic cells (FDCs) that support B cell proliferation in germinal centers (Fig. 4C). CD20+ cell density was significantly higher in CD23+ LAs, both intra-tumoral ($P < 0.001$) and extra-tumoral ($P < 0.001$) (Supplementary Fig. 2D). Together, these data show location-dependent differences in B cell density and the presence of follicular dendritic cells indicative of germinal center-like structures.

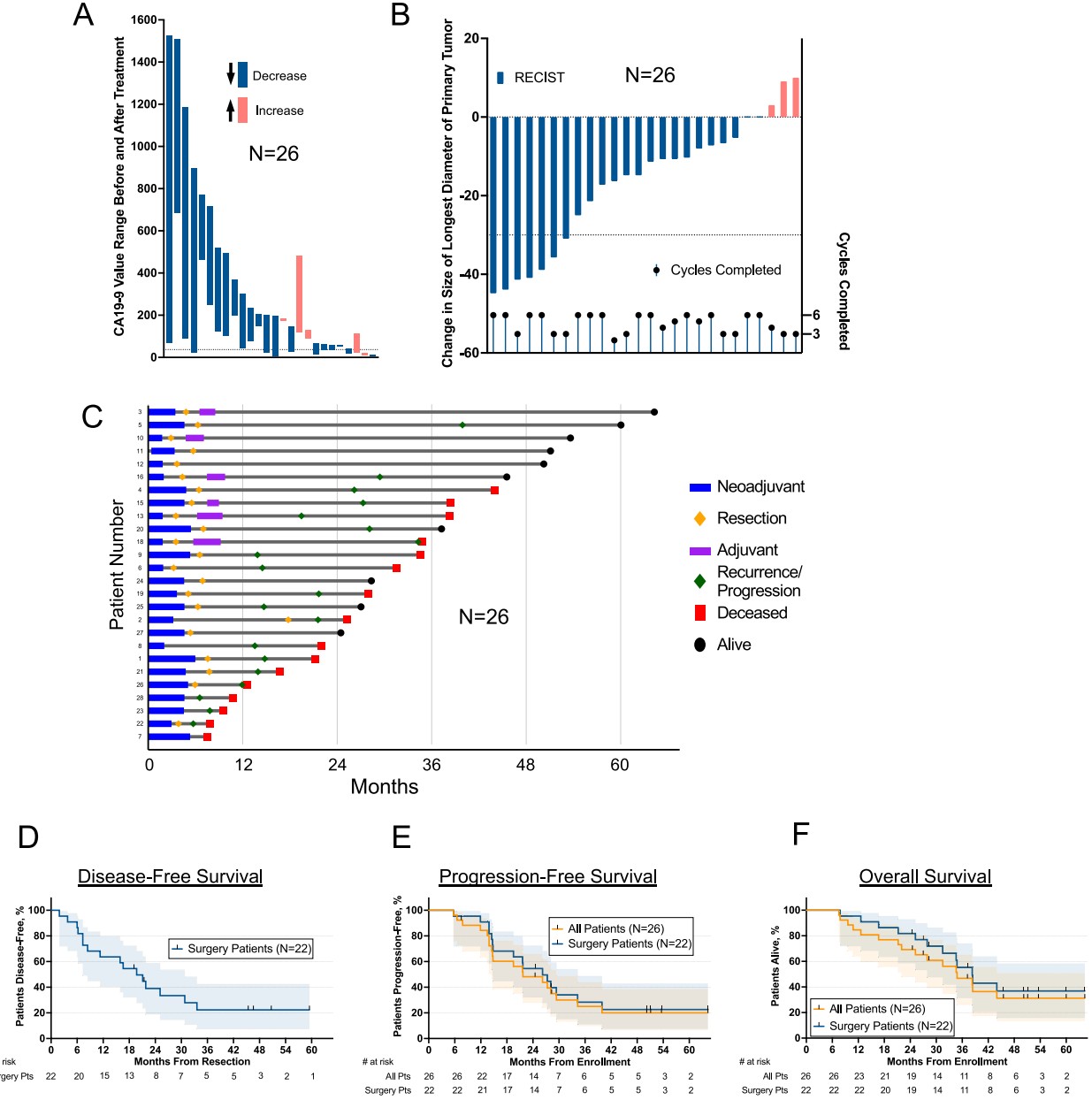

**Fig. 2 | Outcomes for all patients on the trial. A** Blood biomarker CA19-9 changes during the treatment window for all patients treated with FFX and nivolumab. Patients are ranked from left to right on the x-axis by test values on day of first treatment. Post-treatment values are below (blue) or above (orange) pre-treatment values. Dotted line placed at normal/abnormal boundary. **B** RECIST1.1-defined changes in the longest diameter of the primary tumor measured with computed tomography. Dotted line given at threshold for partial response (−30%). The number of cycles completed is given at the bottom and values align with the scale on the lower right y-axis. **C** Swimmers plot for each of the 26 patients. **D** Disease-free survival (time from resection) of the 22 patients treated by surgical resection. **E** Progression-free survival of all 26 patients who completed treatment with accessible medical records and the 22 patients treated by surgical resection. **F** Overall survival of all 26 patients who completed treatment with accessible medical records and the 22 patients treated by surgical resection. For panels **D**–**F**, shaded regions represent 95% confidence intervals. Source data are provided as a Source Data file.

Antigen-reactive B cells can terminally differentiate to PCs in LAs that subsequently migrate throughout the tumor bed[24]. In tumors from trial patients, both the number and density of intra-tumoral PCs significantly correlated with the number and density of intra-tumoral LAs, respectively (Fig. 4D, E). In contrast, intra-tumoral PC number did not correlate with the number of extra-tumoral LAs (Fig. 4D). The number and density of intra-tumoral PCs did not correlate with the number or density of CD20 + B cells in either intra- or extra-tumoral LAs (Supplementary Fig. 2E, F). These findings are consistent with intra-tumoral LAs per se as the primary source of intra-tumoral PCs in tumors treated with mFFX and nivolumab, rather than their capacity to

form GC-like CD20+ clusters (more common in extra-tumoral LAs). In contrast, tumors from patients treated with mFFX alone exhibited significantly fewer intra-tumoral PCs (Fig. 4F) (consistent with Fig. 3E) and no correlation was found between PC density and intra-tumoral LA density in this group (Supplementary Fig. 2G).

**LAs with high PC-to-B Cell ratios are disordered structures with fewer central memory and progenitor exhausted (early effector) T cells**

To better characterize the phenotype of LAs associated with PC production, we performed spatial transcriptomics (10X Genomics Xenium

**Table 3 | Histopathological, Clinical, and Surgical Characteristics of Trial Patients**

| Histopathology | N = 22 | Efficacy Related | N = 22 |
|---|---|---|---|
| Tumor Location | | Pre-treatment characteristics | |
| Head | 11 (50) | Evaluable pretreatment CA19-9 U/mL[b], median [IQR] | 232 [94–536] |
| Head/Uncinate | 1 (5) | | |
| Uncinate | 3 (14) | Tumor size (cm)[c], median [IQR] | 3.0 [2.7–4.0] |
| Body | 3 (14) | Radiographic vascular involvement[d] | |
| Body/Tail | 2 (9) | Venous | 7 (32) |
| No residual tumor | 2 (9) | Arterial | 5 (23) |
| Grade | | Both | 10 (45) |
| G1 - Well differentiated | 1 (5) | Post-treatment, pre-operative characteristics | |
| G2 - Moderately differentiated | 13 (59) | | |
| G3 - Poorly differentiated | 4 (18) | CA19-9, median [IQR] | 37 [20–108] |
| Gx – Unable to assess[a] | 4 (18) | Tumor size (cm), median [IQR] | 2.8 [2.3–3.4] |
| Lymphovascular Invasion | | Radiographic vascular involvement | |
| Present | 6 (27) | Venous | 5 (23) |
| Not identified | 16 (73) | Arterial | 5 (23) |
| Perineural Invasion | | Both | 9 (41) |
| Present | 17 (77) | None | 3 (13) |
| Not identified | 5 (23) | | |
| Resection margins | | Surgical Outcomes | N = 22 |
| R0 | 19 (86) | Type of surgery | |
| R1 | 3 (14) | Whipple | 16 (73) |
| R2 | 0 (0) | Distal pancreatectomy | 5 (23) |
| pT | | Total pancreatectomy | 1 (4) |
| 0 | 2 (9) | Vascular resection | |
| 1 | 8 (36) | Venous resection | 4 (18) |
| 2 | 9 (41) | Arterial resection | 4 (18) |
| 3 | 0 (0) | Both | 2 (9) |
| 4 | 2 (9) | Operative data | |
| x | 1 (5) | Operative time (min), median [IQR] | 422 [342–478] |
| pN | | EBL (mL), median [IQR] | 300 [212–500] |
| 0 | 11 (50) | Postoperative pancreatic fistula[e] – ISPRC definition | |
| 1 | 7 (32) | | |
| 2 | 4 (18) | Grade A, biochemical leak | 1 (5) |
| AJCC stage, 8th edition | | Grade B | 1 (5) |
| 0 | 2 (9) | Grade C | 0 (0) |
| 1 | 9 (41) | Other Complications | |
| 2 | 5 (23) | Clavien-Dindo grade≥3, within 90 days | 6 (27) |
| 3 | 6 (27) | | |
| CAP score | | | |
| 0 - Complete response | 2 (9) | | |
| 1 - Near complete response | 2 (9) | | |
| 2 - Partial response | 16 (72) | | |
| 3 - Poor or no response | 2 (9) | | |

[a]Grade was unable to be assessed in complete responders with no residual tumor, as well as near-complete responders with only few scattered or small clusters of tumor cells remaining.

[b]CA19-9 was evaluable and included only if total bilirubin at the time of blood draw was <3 mg/dL. [c]Tumor size was measured on computed tomography (CT) scan. [d]Vascular involvement was identified on CT scan and includes the celiac artery, common hepatic artery, superior mesenteric artery, portal vein, and superior mesenteric vein. [e]Postoperative pancreatic fistula as defined by the International Study Group of Pancreatic Fistula (ISGPS). *AJCC* American Joint Committee on Cancer, *CAP* College of American Pathologists, *EBL* estimated blood loss.

5 K Prime) on six resected, classical-subtype tumors from trial patients. Unbiased single-cell clustering identified 20 PDAC-associated cell types that were mapped back to their position in the tissue section and superimposed on an H&E stain for pathology review (Supplementary Fig. 3A, B). Given that our IHC analysis showed no positive correlation between CD20 density in LAs and intra-tumoral PC density (Supplementary Fig. 2E), we developed a B cell-independent approach to demarcate LAs by co-occurrence of CCL19 and CXCL13 expressing cells (both cytokines promote LA formation by non-redundant mechanisms[25,27] (Fig. 5A).

Across all 6 tumors, 67 LAs were identified (median of 711 cells). Fibroblasts (16%) and myeloid cells (13%) were present, but T cells, B cells, or PCs comprised >50% of cells (Fig. 5B) and were among the most enriched populations compared with surrounding tumor (Fig. 5C), though enrichment varied among patients (Supplementary Fig. 3C). LAs with high T and B cell densities were significantly larger than those with lower densities (Supplementary Fig. 3D), but neither B nor T cell density correlated positively with PC density (Supplementary Fig. 3E). Notably, 51% of LAs contained more PCs than B cells (high PC-to-B cell ratio, PBR), which were unevenly distributed across patients (Supplementary Fig. 3F).

To assess T cell composition we clustered LA T cells and identified 9 distinct populations: innate T cells ($\gamma\delta$T, NK T), progenitor exhausted ($T_{PEX}$, GZMK + ), terminal exhausted ($T_{TEX}$, GZMB + PRF1 + ), central memory ($T_{CM}$), resident memory ($T_{RM}$), regulatory ($T_{REG}$), follicular helper ($T_{FH}$) CD4 + T cells, LA organization-related ($T_{ORG}$, defined best by transcript bleed over from adjacent cells expressing CXCL9, CCL19, and CCL21), and proliferating ($T_{PRO}$))(Supplementary Fig. 4A, B). $T_{CM}$, $T_{FH}$, and $T_{ORG}$ cells were most enriched in area inside LAs over outside LAs (Supplementary Fig. 4E). Conventional (low PBR) LAs were enriched for $T_{CM}$ and $T_{PEX}$ cells, whereas high PBR LAs were enriched for $T_{TEX}$ cells (Supplementary Fig. 4F). Accordingly, for all LAs, the PBR positively correlated with the density of $T_{TEX}$ cells ($P = 0.011$), negatively correlated with the density of $T_{PEX}$ cells ($P < 0.001$), and positively correlated with the ratio of $T_{TEX}$ to $T_{PEX}$ densities ($P < 0.001$) (Fig. 5D, E and Supplementary Data). Similar significant relationships were identified between the PBR and non-cytolytic T cell subsets (Supplementary Fig. 4G and Supplementary Data). For the 6 patients analyzed, the PBR-related densities of $T_{TEX}$ and $T_{PEX}$ cells in LAs matched their densities in areas outside LAs ($P = 0.058$) (Fig. 5F), highlighting the potential impact of PBR on T cell function throughout the tumor microenvironment. These findings suggest that the PBR of intra-tumoral LAs in this study is connected to production of terminally exhausted T cell states.

## Discussion

PD-1 inhibition has not been previously evaluated in combination with standard-of-care mFFX as neoadjuvant therapy for patients with borderline resectable pancreatic cancer (BRPC) – a setting in which mFFX and surgery achieve a nearly 2-year median overall survival[13,28]. This clinical context provides a compelling rationale for testing whether neoadjuvant PD-1 blockade can confer clinical benefit beyond chemotherapy while promoting favorable anti-tumor immune cell changes. In this study, we identified four major findings: (1) neoadjuvant PD-1 blockade did not lead to apparent treatment response or survival advantage beyond mFFX; (2) nivolumab was associated with parallel increases in intra-tumoral CD8 T cells and plasma cells; (3) plasma-cell enrichment was linked to increased number and density of intra-tumoral, not extra-tumoral lymphoid aggregates (LA); and (4) more mature intra-tumoral LAs with lower plasma cell-to-B cell ratios (PBR) contained more favorable anti-tumor T cell populations than high PBR LAs.

Although, overall survival was not improved in the cohort as a whole, several notable exceptions were observed. Two patients, including one with Lynch syndrome (loss of MLH1 and PMS2), achieved

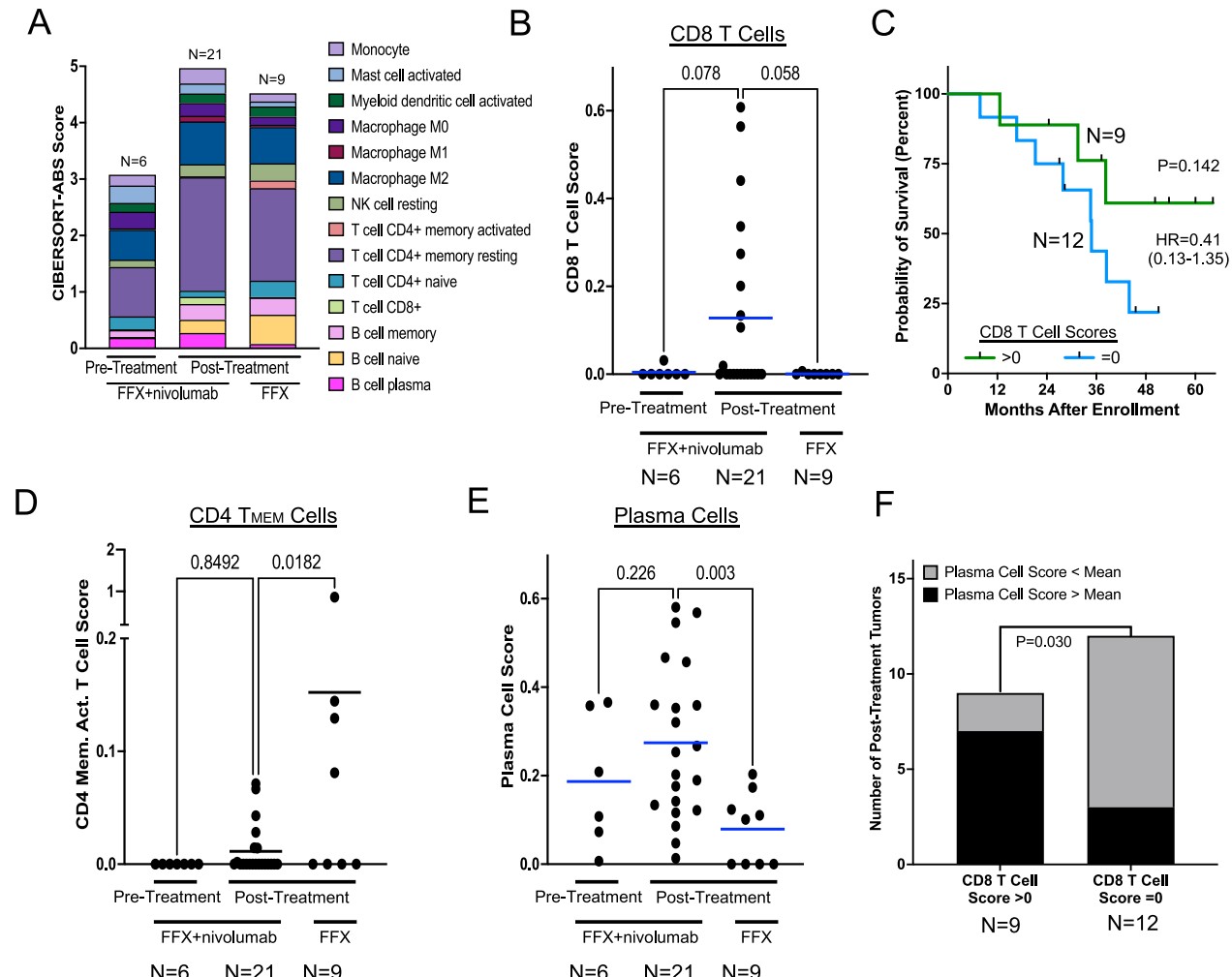

**Fig. 3 | Leukocyte composition within pre- and post-treatment tumors.**
**A** CIBERSORT-ABS estimated abundances of leukocytes from pre-treatment biopsies and post-treatment resection specimens from patients treated with FFX and nivolumab and resection specimens from patients treated with FFX alone. Only leukocyte types with mean scores >0.1 for any cohort are shown. Pre-treatment leukocytes and lymphocytes (B cells, T cells, and plasma cells) from trial patients were more abundant than post-treatment samples from the clinical trial by Mann-Whitney tests (P = 0.028 and P = 0.065, respectively). **B** Individual tumor CIBER-SORT scores for CD8 T cell abundance from pre-treatment biopsies and post-treatment resection specimens from patients treated with FFX and nivolumab and resection specimens from patients treated with FFX alone. Differences in means tested by one-way ANOVA with uncorrected Fisher's LSD. **C** Kaplan Meier plot of overall survival of patients with post-treatment CIBERSORT-ABS CD8 T cell scores >0 (green line) versus scores of 0 (blue line). **D** Individual tumor CIBERSORT scores for CD4 memory activated T cells or **E** plasma cell abundance from pre-treatment biopsies and post-treatment resection specimens from patients treated with FFX and nivolumab and resection specimens from patients treated with FFX alone. Differences in means tested by one-way ANOVA with uncorrected Fisher's LSD. **F** Tumors treated with FFX and nivolumab with measurable T cell scores were significantly more likely to have plasma cell scores >mean (P = 0.030, Fishers Exact test, two-tailed). Source data are provided as a Source Data file.

complete pathologic responses (CAP score 0), an outcome rarely seen with mFFX alone. Another patient had a near complete response (CAP score 1) and has remained disease-free for 4 years. Three additional patients with partial responses (CAP score 2), including one with LN metastases, are also recurrence-free after four years. Together, these outcomes suggest a 21% (6 of 28) long term survival rate, which compares favorably with other BRPC studies[29]. These durable responses in a minority of patients highlight the potential existence of a yet-to-be-defined PDAC subset that benefits from PD-1 blockade plus mFFX and underscore the need for predictive biomarkers beyond MSI status and PD-1/PD-L1 expression. Our study was limited by insufficient powering for efficacy analyses beyond pathologic outcomes and did not include a comparator arm for mFFX treatment alone. For molecular analyses, we were restricted to resected, post-treatment tumor tissue and 6 pre-treatment samples; we did not assess blood or lymph nodes, important sites in the cancer-immunity cycle. However, based on some of the data reported herein, a randomized trial comparing

chemotherapy to chemoimmunotherapy in BRPC has been launched (JCOG1908E).

Consistent with the known effects of anti-PD-1, we observed that the addition of nivolumab to mFFX modestly increased intra-tumoral CD8 T cells and plasma cells, which tended to rise together compared to stage-matched historical controls treated with mFFX alone. Patients with higher intra-tumoral CD8+ T cell densities had longer median overall survival, though the association was modest. In other solid tumors, clinical responses to anti-PD-1 therapy are associated with clonal expansion and functional reinvigoration of tumor-reactive T cells[17]. In PDAC, anti-PD-1 therapy has been shown to induce proliferation of likely tumor-reactive effector CD8+ T cells accompanied by strong NFκB-driven transcriptional programs – a signature associated with nonproductive T cell responses in this context[30]. In contrast, productive immune responses to PD-1 blockade in other GI cancers feature robust IFNγ signaling without concurrent NFκB activation, a pattern characteristic of responders in the broader GI cancer setting[31].

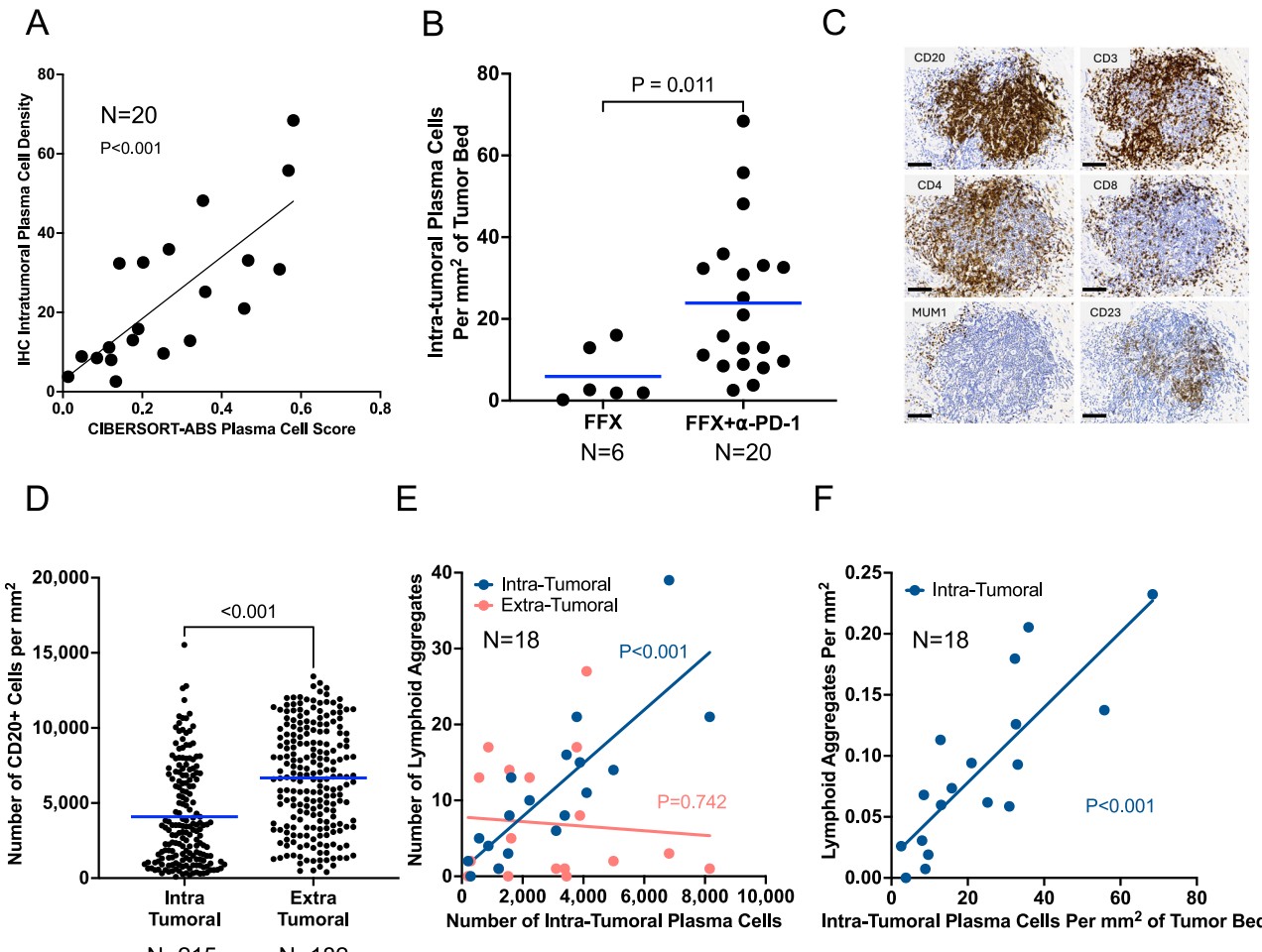

**Fig. 4 | Immunohistochemistry analyses of lymphoid aggregates from resected tumors post-treatment. A** Agreement between plasma cell scores from CIBER-SORT analysis and intra-tumoral plasma cell density determined by IHC. **B** Tumors from patients treated with FOLFIRINOX alone have significantly lower intra-tumoral plasma cell density than those treated with FOLFIRINOX and nivolumab ($P = 0.011$, two-tailed Mann-Whitney test). **C** An example lymphoid aggregate from immuno-histochemistry to demonstrate the technique performed on serial sections to identify cells expressing CD20, CD3, CD4, CD8, CD20, CD23/FCER2, and MUM1/ IRF4. Scale bars represent 100 μm. **D** Intra-tumoral LAs have significantly lower CD20+ cell density than extra-tumoral LAs ($P < 0.001$, two-tailed Mann-Whitney test). **E** The number of intra-tumoral plasma cells positively correlates with the number of intra-tumoral ($P < 0.001$) but not extra-tumoral lymphoid aggregates. **F** The density of intra-tumoral plasma cells positively correlates with the density of intra-tumoral lymphoid aggregates ($P < 0.001$). $P$ values for panels **A**, **E**, **F** derived from simple linear regression. Source data are provided as a Source Data file.

Our findings suggest that a defect within intra-tumoral LAs may contribute to the emergence of dysfunctional T cell phenotypes in nivolumab-treated PDAC tumors, potentially limiting treatment efficacy.

We observed that plasma cell enrichment was associated with an increase in the number and density of intra-tumoral but not extra-tumoral LAs. LAs in PDAC may be either mature or immature. Mature LAs, defined by size, B cell density, and the presence of CD23+ follicular dendritic cells (FDCs), possess TLS and germinal center features typically linked to better outcomes[32-34]. Unlike conventional approaches that define LAs based on their expected cellular composition, we used image-based, single-cell spatial transcriptomics to identify them based on focal expression of key lymphocyte-attracting chemokines. This method identified canonical mature TLSs in anti-PD-1-treated tumors and revealed a large population of irregular LAs with low B cell density and abundant PCs.

Recent pan-cancer analyses of humoral immune cells have identified subsets of PCs that likely arise from antigen-experienced B cells via an extrafollicular (EF), non-germinal center route[35,36]. EF-derived PCs exhibit lower somatic hypermutation and class switching, secrete low-affinity antibodies, and are enriched in tumor types such as PDAC,

a profile associated with poor clinical outcomes[36]. In contrast, their B cell precursors have enhanced antigen-presenting capacity, respond to IL-21 from $T_{FH}$ cells, and are enriched in immunotherapy-responsive tumors such as melanoma and NSCLC[35]. This dichotomy aligns with our finding that many intra-tumoral LAs in PDAC after nivolumab had high PC-to-B cell ratios—a pattern that is suggestive of B cell depletion and premature differentiation into short-lived EF-like PCs. In our study, we also observed a nivolumab-associated increase in PCs, paralleling findings in NSCLC and melanoma, where PC infiltration has been associated with ICI responsiveness[26,37]; however, those studies did not report B cell depletion or evidence of extrafollicular differentiation. These differences highlight the context-specific nature of humoral responses to PD-1 blockade and support the interpretation that, in PDAC, PC enrichment may reflect a dysfunctional rather than productive immune response.

Anti-PD-1 therapy reverses PD-L1-mediated suppression of PD-1 expressing, tumor antigen-experienced CD8+ T cells, counteracting a key mechanism of immune resistance[38]. In parallel, anti-PD-1 antibodies also act on PD-1+ CD4+ $T_{FH}$ cells in tumor-draining lymph nodes, promoting CD8+ T cell expansion and contributing to treatment efficacy in mouse models of melanoma and colon cancer[39,40]. However,

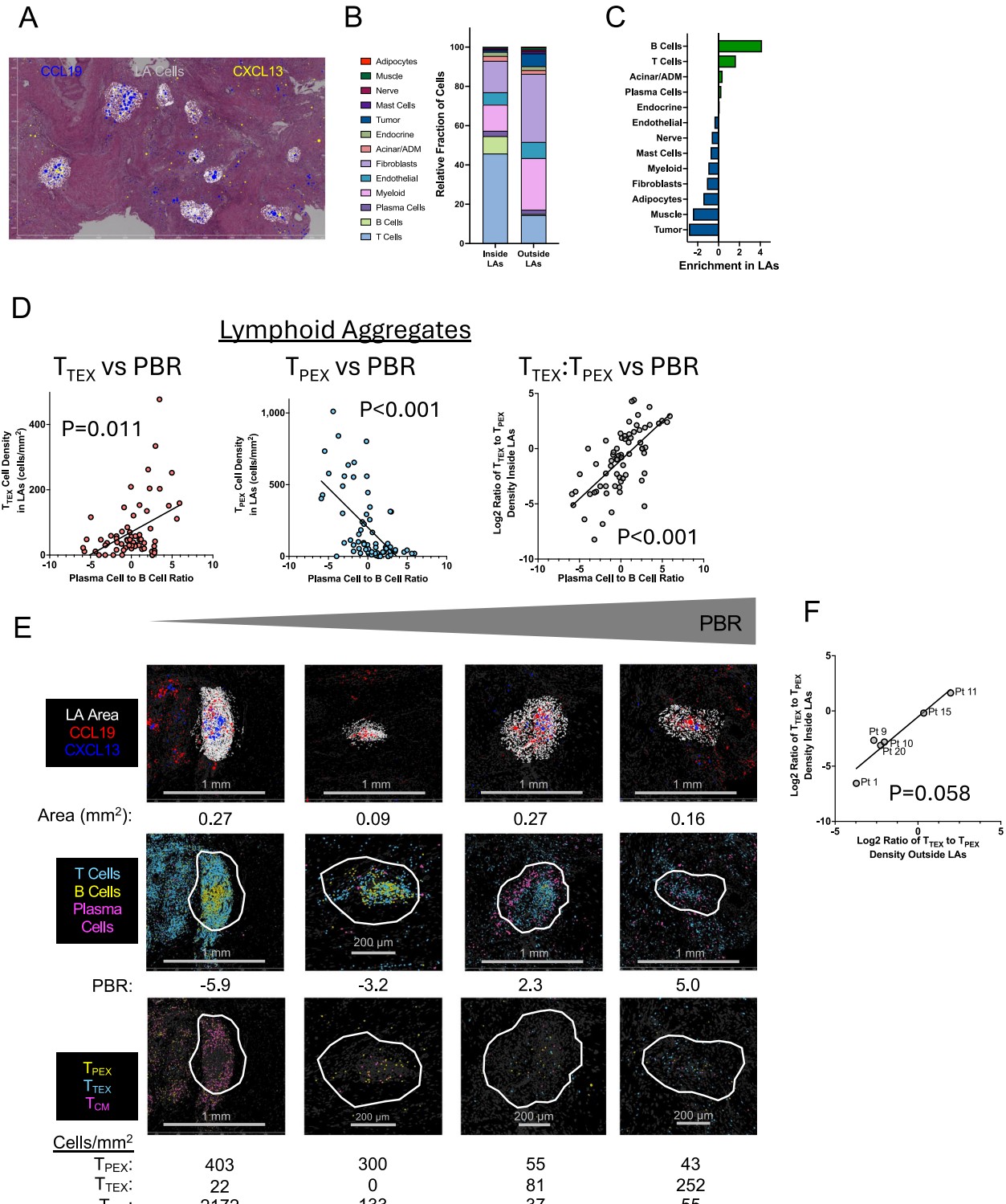

**Fig. 5 | Image-based spatial transcriptomics assessment of lymphoid aggregates from 6 trial patient resected tumors. A** An example image showing the technical use of CCL19 (red) and CXCL13 (blue) transcripts to define LAs (cells shaded white). Distance between scale bar tics is 500 μm. **B** The mean relative fraction of each cell type inside all 67 LAs and in tissue outside of LAs. The 4 tumor, 2 fibroblast, and 3 T cell clusters were each combined into single clusters. **C** Relative enrichment of each cell type inside LAs. **D** Correlation of the PBR to densities of cytolytic T cell subsets (left and middle plots) or the LOG$_2$ ratio of $T_{TEX}$ to $T_{PEX}$ densities (right plot). Trend lines plotted from simple linear regression, P values

from Spearman non-parametric correlation. Each plot displays data from 67 LAs. **E** Spatial transcriptomics images of 4 representative LAs demarcated by CCL19/CXCL13 expression (top row), LA lymphocytes (middle row), and LA T cell subsets (bottom row). LAs are ordered from left to right by increasing PBR. The area, PBR, and T cell subset densities are given for each LA. **F** Correlation between the LOG$_2$ ratio of $T_{TEX}$ to $T_{PEX}$ densities inside versus outside LAs for 6 patients. Trend line plotted from simple linear regression, P value from Spearman non-parametric correlation. Source data are provided as a Source Data file.

PD-1 signaling plays a critical role in supporting $T_{FH}$ production of IL-21, which drives B cell proliferation and differentiation. Disruption of this pathway impairs the generation of memory B cells and long-lived PCs in both mice and humans[41,42]. Taken together, these mechanistic insights may help explain our fourth observation: more mature intra-tumoral LAs with lower PC-to-B cell ratios also contained higher proportions of T cell subsets with effector phenotypes, whereas high-PBR LAs were enriched for terminally exhausted T cells and PCs. This pattern suggests that in PDAC, PD-1 blockade may inadvertently disrupt productive B cell responses, driving premature plasma cell differentiation and T cell exhaustion, thereby limiting clinical benefit.

Our spatial transcriptomic analysis revealed that anti-PD-1-driven LA irregularities extended to the T cell compartment. High PBR LAs were depleted of $CD4^+$ $T_{CM}$ and $CD8^+$ $T_{PEX}$ cells – key populations associated with responses to anti-PD-1 therapy[43–45]—and were enriched for CD8 + $T_{TEX}$ cells. Accordingly, the density of $T_{TEX}$ and $T_{PEX}$ cells and their ratio significantly correlated with PBR across all LAs analyzed (Fig. 5D, E), and the ratio of $T_{TEX}$ to $T_{PEX}$ cells matched between intra-LA and extra-LA regions for the 6 tumors analyzed (Fig. 5F), demonstrating the contribution of high PBR LAs to the PDAC TME in general. Notably, the reduction in $T_{CM}$ cells within high PBR LAs paralleled the nivolumab-associated decrease in activated memory $CD4^+$ T cells identified by bulk transcriptomic analysis (Fig. 3D). Taken together, lymphocyte subsets with the potential to expand as effector cells ($T_{CM}$ cells and most B cells) were enriched in conventional, low PBR LAs, whereas terminally differentiated effector lymphocytes ($T_{PRO}$, $T_{TEX}$, and PCs) were associated with irregular, high PBR LAs.

In microsatellite-stable PDAC, where tumor-reactive $CD8^+$ T cells are scarce, effective tumor control may depend more heavily on coordinated lymphocyte activity within LAs than in other solid tumors. Our findings suggest that anti-PD-1 therapy impairs LA function in PDAC by skewing B cell differentiation and depleting key T cell subsets. LA irregularity may be compounded by tumor-intrinsic factors, as it is accentuated in LAs within the tumor bed compared to those in adjacent tissue. Taken together, these findings suggest that in PDAC, the limited efficacy of anti-PD-1 therapy may stem not only from a paucity of pre-existing, tumor-reactive T cells, but also from therapy-induced disruption of intra-tumoral LA structure and function. Therapeutic strategies that counteract these deleterious effects – by preserving or restoring the integrity of the adaptive immune response in LAs – may be essential to unlocking the full potential of immunotherapy in PDAC.

## Methods

### Inclusion and Ethics
This neoadjuvant trial was reviewed and approved by an ethical review board (UCLA Institutional Review Board) and was conducted in accordance with the ICH Harmonized Tripartite Guideline for Good Clinical Practice and the principles of the Declaration of Helsinki. The study was monitored by the UCLA Data Safety Monitoring Board (DSMB) in accordance with institutional policy. All patients provided written informed consent.

### Study Design
This single-arm, prospective, open-label Phase 1 pre-operative investigator-initiated study (Clinical Trials.gov identifier NCT03970252, registration date 5/31/2019) enrolled adults with biopsy-proven borderline-resectable pancreatic ductal adenocarcinoma (BRPC) at two academic hospitals between August 2019 and March 2023. Participants received up to six 28-day cycles of modified FOLFIRINOX (oxaliplatin 85 mg/m², leucovorin 400 mg/m², irinotecan 150 mg/m², and 5-fluorouracil 2400 mg/m² by 46-h infusion) combined with nivolumab 480 mg intravenously on day 1 of each cycle. Treatment was continued for either 6 cycles (3 months) up to a maximum of 12 cycles (6 months) of treatment. The decision between 3 or 6 months of pre-operative treatment was investigator discretion after a multi-disciplinary tumor board discussion. Because this regimen had been explored in advanced disease, no formal dose escalation was performed in this pre-operative study. Patients without progression proceeded to surgery 4-6 weeks after their last dose. The pre-specified primary endpoints were safety and tolerability with a particular focus on post-operative complications (pancreas fistulas as defined in the protocol). The primary objectives were safety (defined by the frequency of grade ≥3 treatment-related adverse events (AEs) per CTCAE v5.0) and feasibility (defined as the proportion of patients taken to surgical resection). Secondary objectives included radiographic and biochemical response rates, R0 resection rate, and disease-free survival (DFS). After the first 6 patients were enrolled, the safety of the patients was reviewed by the DSMB before proceeding to the second part of the study. This study was preregistered on ClinicalTrials.gov (NCT03970252) on May 31, 2019. All primary and secondary endpoints and safety monitoring procedures were prespecified in the pre-registered protocol. Post hoc exploratory translational analyses were conducted to further characterize treatment-associated immune features and did not influence study conduct or endpoint assessment.

### Patient Population
Eligible patients were patients >18 years diagnosed with Borderline Resectable Pancreatic Cancer (BRPC) as per National Comprehensive Cancer Network (NCCN) Criteria. All patients were reviewed by a qualified radiologist, surgeon, and medical oncologist at a multi-disciplinary tumor board to confirm BRPC. Patients were required to have a cytologic or histologic diagnosis of Pancreatic Ductal Adenocarcinoma (PDAC). Patients were required to have an Eastern Cooperative Oncology Group (ECOG) performance status of 0 or 1 and adequate renal, hepatic and hematology lab values. The key exclusion criteria included any prior treatment (including immunotherapy) for PDAC, a diagnosis of locally advanced or metastatic PDAC, history of active autoimmune conditions, and conditions requiring >10 mg prednisone or equivalent. Complete eligibility parameters are described in the study protocol (available in the Supplementary Information).

### Clinical Assessment
Baseline evaluation included contrast-enhanced CT of the chest/abdomen/pelvis, serum CA19-9, complete blood count, comprehensive metabolic panel, and ECOG scoring. CT imaging was repeated every twelve weeks (after cycles 3 and 6) and interpreted according to RECIST v1.1. Treatment-emergent AEs were recorded at each visit and graded with CTCAE v5.0. Surgical eligibility required stable or improved disease on restaging, absence of distant metastasis, and vascular involvement amenable to venous reconstruction, if necessary. Resected specimens were evaluated by gastrointestinal pathologists for margin status and treatment effect using the College of American Pathologists 0–3 scale.

Post-resection follow-up visits occurred every three months for the first two years and every six months thereafter, including CT imaging and CA19-9 testing. DFS was calculated from resection to recurrence; PFS from enrollment to recurrence; OS from enrollment to death from any cause. Patients without an event were censored at the time of data cut-off (Q1 2025).

### Statistical Analysis and Study Endpoints
The safety population consisted of all patients receiving at least one dose of the study treatment. The efficacy population consisted of all patients in whom pathologic response was available. The primary endpoint of the study was safety and tolerability with special attention to rates of post-surgery complications such as pancreatic fistulas using the 2016 International Study Group (ISGPS) definition and grading schema[46], increased rates of other post-operative complications, and delays in surgery. A co-primary endpoint was pathologic complete response as measured using the College of American Pathology (CAP)

treatment response scoring system. Pre-specified secondary end-points included percent change of CA 19-9, radiographic response rate, R0 resection rate and disease-free survival (DFS). Exploratory analyses included progression free survival (PFS), and overall survival (OS). DFS was defined as time since resection until disease recurrence or disease-related death. PFS was defined as time since enrollment until disease-recurrence or disease-related death. OS was defined as time since enrollment until death from any cause. Survival endpoints were analyzed using Kaplan-Meier point estimates. Data were collected at University of California, Los Angeles between March 2022 and December 2024. Statistical analyses were calculated using GraphPad Prism. All patients were monitored for 120 days post-surgery for the emergence of any post-operative complications. Safety was monitored by the number and grade of adverse events (AEs) or immune-related AEs using the Common Terminology Criteria for Adverse Events (CTCAE), version 4.0[47]. AEs leading to surgical delay of >2 weeks were reported and post-operative AEs were graded according to Clavien-Dindo grading of surgical complications[48].

## Specimen Processing

Pre-treatment diagnostic biopsies and surgical resection specimens were obtained from the UCLA Translational Pathology Core Laboratory or from non-UCLA sites where biopsies were performed. Specimens were processed through standard diagnostic pathology workflows and access to remnant pathologic material was approved by the UCLA Institutional Review Board protocol (IRB-11-2112). Surgical pathology cases were reviewed by a practicing pathologist (DD) who selected appropriate formalin-fixed paraffin-embedded (FFPE) tissue blocks from which standard H&E stained and serial unstained histologic sections for downstream analysis were generated by the UCLA Translational Pathology Core Facility.

## RNA Sequencing

H&E-stained sections from pre-treatment biopsies and resection specimens were screened by a board-certified subspecialty gastrointestinal pathologist (DD) to designate regions of tumor that were further enriched through macrodissection. The indicated region was scraped from 10 μm unstained sections on each of 10 slides. RNA extraction and sequencing were performed by The Center for Genomics and Biotechnology at UCLA. Libraries for RNA-Seq were prepared with KAPA Stranded RNA-Seq Kit with RiboErase Kit. Sequencing was performed on Illumina NovaSeq 6000 for PE 2×50 run. Data quality check was performed on Illumina SAV. Demultiplexing was performed with Illumina Bcl2fastq v2.19.1.403 software. Alignment was performed using STAR[49] with human reference genome GRCh38. The Ensembl Transcripts release GRCh38.108 GTF was used for gene feature annotation. CPM normalized counts were generated by adding 1.0E-4 followed by counts per million (CPM) normalization.

## Analysis of RNA-seq Gene Expression

CIBERSORT-ABS was performed on CPM-normalized RNA-seq data using TIMER2.0[50]. Tumor cell intrinsic (PurIST) and CAF-intrinsic (DeCAF) subtyping was performed as described[19,51]. The following pairs of genes were used for PurIST as previously described[52]: GPR87/REG4, KRT6A/ANAX10, KRT17/LGALS4, S100A2/TFF1, C16orf74/DDC, KRT15/PLA2G10, PTGES/CDH17, and DCBLD2/TSPAN8. For the post-treatment tumor from patient 22, RNA-seq was performed on 2 independent samples with similar results; the CIBERSORT-ABS, PurIST, and DeCAF scores were averaged from the 2 samples to arrive at a single value for the tumor.

## Immunohistochemical Staining of Human Samples

Antibody validation and IHC staining of FFPE tissue sections was performed by the UCLA Translational Pathology Core Laboratory (Supplementary Table 1). Sections were deparaffinized and rehydrated before undergoing antigen-retrieval and washing steps. Rabbit or mouse primary antibodies (Supplementary Table 1) were diluted, applied, and incubated overnight (4 °C) prior to incubation with HRP for 10 min. DAB solution with hematoxylin counterstain was applied on the sections for 10 min, dehydrated, mounted, and left to dry. H&E-stained slides were imaged at 40× with a Motic EasyScan One slide scanner. QuPath software v0.5.1 was used to identify and enumerate cells and calculate cell densities.

## Lymphoid Aggregate Characterization

Lymphoid aggregates (LAs) were identified by non-random appearing, compact clusters of >50 CD20+ cells associated with proximal compact clusters of CD3+ cells. QuPath was used to outline selected LAs, calculate their area and measure cells expressing CD20, CD3, CD23, and MUM1 on serial sections. The density of each cell type within each LA was calculated and oriented in relation to the tumor bed annotated by the gastrointestinal pathologist. Intra-tumoral plasma cell density was calculated by the number of MUM1-positive cells within the tumor bed margin.

## Spatial Transcriptomics

For 10x Genomics Xenium spatial transcriptomics, 5 μm sections of FFPE tumor tissue from each patient were placed within the fiduciary frame of Xenium slides, with tissue from three patients processed per slide. Xenium was performed by the UCLA Technology Center for Genomics and Bioinformatics and data were processed by the Xenium Onboard Analyzer. Xenium spatial transcriptomics data was processed using a computational pipeline implemented in R with the Seurat framework[53]. Samples were processed by extracting gene expression matrices (including 5,101 genes) from cell feature matrix H5 files. Preliminary Xenium Analyzer on-board, graph-based clustering was performed independently on each sample and up to 500 representative cells were selected from each cluster from each sample with expression profiles most similar to the mean expression of their originally assigned cluster. Cells were filtered to include only those with at least 10 detected genes and 20 total transcripts. The merged dataset was normalized, dimensionally reduced with PCA, and clustered using the Louvain algorithm at a resolution of 0.2 to identify 20 distinct clusters. UMAPs were also generated in Seurat.

Clusters were manually annotated based on marker gene expression specific to each cluster and position on the Xenium image. Cell types were validated by a pathologist based on a superimposed H&E image. All cells from each of the 6 samples were assigned to clusters using Seurat's label transfer workflow. Cells with prediction scores below 0.6 (5–10% of each sample) were designated as "unassigned" and not used for further analyses.

Lymphoid aggregates were defined by selecting all CCL19-positive cells within 50 μm of a CXCL13-positive cell and identifying 15 or more of these cells within a 75 μm radius using DBSCAN. Clusters with centroids within 400 μm were merged, and concave hull polygons with a 100 μm buffer were generated around each cluster to define LA boundaries. Cells within the polygon boundary were assigned to the LA.

Cells from the CD4 + , CD8+ and mitotic (predominantly T cells) clusters were extracted from all samples and T cell gene expression matrices were created from the original H5 files, merged, and processed using the Seurat workflow. High-resolution clustering was used to identify and remove non-T cells. After a final round of clustering, eight distinct T cell populations were manually annotated based on cluster-specific gene expression. Cell assignments were exported for each sample and integrated with LA spatial data for downstream analyses (Supplementary Data).

## Data Analysis and Statistics

All computations were performed with R (version 2025) or Microsoft Excel (version 16) and all statistical tests were performed using

GraphPad Prism (version 10). Cell type enrichment scores inside LAs or in low PBR LAs were determined by the $LOG_2$ ratio of the relative fractions of the cell types. The plasma cell to B cell ratio was calculated as the difference between the $LOG_2$(number of PCs +1) and $LOG_2$(number of B cells +1).

## Reporting summary

Further information on research design is available in the Nature Portfolio Reporting Summary linked to this article.

## Data availability

The RNA-seq data generated in this study have been deposited in the Gene Expression Omnibus under accession number GSE313101. The spatial transcriptomics data generated in this study have been deposited in the Gene Expression Omnibus under accession number GSE313662. A redacted study protocol is provided in the Supplementary Information. The remaining data supporting the findings of this study are available in the article, its supplementary information files, or the source data provided with this paper. Source data are provided with this paper.

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

## Acknowledgements

This work was funded by NCI RO1CA250529, the Jonsson Comprehensive Cancer Center, and The Hirshberg Foundation for Pancreatic Cancer Research. We thank Bristol Myers Squibb for providing nivolumab and some financial support for this clinical trial. Expert technical support was provided by the Translational Research Pathology Core and the Technology Center for Genomics & Bioinformatics at UCLA.

## Author contributions

Z.A.W. and T.D. designed the clinical trial, enrolled patients, evaluated safety and endpoints, and contributed to manuscript writing, review, and editing. J.M.L. collected clinical data, designed experiments, generated data and figures, and wrote the manuscript. D.W.D. obtained resources and contributed to manuscript writing, review, and editing. S.Z. collected clinical data, designed experiments, generated data and figures, and contributed to manuscript writing and editing. A.P., M.S., L.L., E.R.A., and C.G.R. participated in critical discussions and manuscript review and editing. L.R., S.K., J.K., O.J.H., M.G., S.S., O.O., D.W., L.Y., A.M.S., K.K., and C.K. evaluated patients on the trial. M.H., S.E.K, O.T., Z.L. collected clinical data. C.-H.T. performed statistical analyses.

## Competing interests

Dr. Wainberg is an advisor for Amgen, Alligator, Astra Zeneca, Daiichi, Bayer, BMS, Merck, Ipsen, Gilead, Arcus, Astellas, Lilly, Pfizer, Revolution Medicine. Dr. Wong reports consulting from Merck, RAPT Therapeutics, Coherus Biosciences, Fennec Pharma, Merus and Bicara. Drs. Radu and Donahue report support from Trethera Corporation outside the submitted work. The other authors declare no competing interests.

## Additional information

Zev A. Wainberg ◉ [1,2,3,7] ✉, Jason M. Link ◉ [2,3,7] ✉, Alykhan Premji[3], Serena Zheng ◉ [3], Michael Srienc ◉ [3], McKensie Hammons[4], Shineui E. Kim[4], Luyi Li[3], Zeyu Liu[4], Olga Tsvetkova[4], Evan R. Abt[2,5], Lee Rosen[1], Stephen Kim[1], Jonathan King[3], O. Joe Hines[2,3], Mark Girgis[2,3], Saeed Sadeghi[1], Olga Olevsky[1], Deborah Wong[1], Lisa Yonemoto[1],

**Ann Marie Siney[1], Kim Kelly[2], Christine Kivork[2], Chi-Hong Tseng ⓘ [1], Caius G. Radu ⓘ [2,5], David W. Dawson[2,6] & Timothy R. Donahue ⓘ [2,3,5] ✉**

[1]Department of Medicine, David Geffen School of Medicine at University of California Los Angeles, Los Angeles, CA, USA. [2]Jonsson Comprehensive Cancer Center, University of California Los Angeles, Los Angeles, CA, USA. [3]Department of Surgery, University of California Los Angeles, Los Angeles, CA, USA. [4]David Geffen School of Medicine, University of California Los Angeles, Los Angeles, CA, USA. [5]Department of Molecular and Medical Pharmacology, University of California Los Angeles, Los Angeles, CA, USA. [6]Department of Pathology and Laboratory Medicine, University of California, Los Angeles, CA, USA. [7]These authors contributed equally: Zev A. Wainberg, Jason M. Link. ✉e-mail: zwainberg@mednet.ucla.edu; jmlink@mednet.ucla.edu; tdonahue@mednet.ucla.edu

