## [Transparent Peer Review file · Nature Communications]

Neoadjuvant modified FOLFIRINOX plus nivolumab in borderline-resectable pancreatic ductal adenocarcinoma: a pilot phase 1 trial

Corresponding Author: Dr Timothy Donahue

Version 0:

Reviewer comments:

Reviewer #1

(Remarks to the Author)

This is an interesting study looking into the effects of PD-1 blockade with FOLFIRINOX in patients with pancreatic cancer receiving neoadjuvant therapy. The strength of the study is the detailed examination of the resected specimens. The authors have convincingly responded to the reviewers comments. I recommend publication of the article and have no further comments.

Reviewer #2

(Remarks to the Author)

This manuscript summarized a pilot study (based on the protocol). The data analysis is mainly descriptive for the primary and secondary outcomes, with several exploratory analysis reported significant associations. Overall, the report is generally well written with several minor issue to be addressed:

The title should state as a pilot trial rather than " A Phase 1 Trial".

The protocol stated that the trial "will be conducted in two parts: where in Part I (6-12 patient safety lead in period). Please briefly describe the results from the part I results in the method. Also, per protocol, "Two interim safety assessments will be performed when first 7 and 15 evaluable patients are enrolled in the study", please also briefly describe the results. There is sample size justification in the protocol, please consider to add it in the method.

Reviewer #4

(Remarks to the Author)

This paper presents the results of a phase 1b/2 trial evaluating neoadjuvant modified FOLFIRINOX plus nivolumab in patients with borderline-resectable PDAC. A deep characterization of the immune microenvironment was performed and lymphoid aggregates containing high amount of plasma cells were described in patients receiving FOLFIRINOX plus nivolumab. Overall, the authors generated a valuable dataset using cutting edge methods and derived observations that could inform immunotherapy developments.

1) Concerning the title: The authors use the term "Irregular" [...] Lymphoid Aggregates". Usage of the vague term "irregular" contrasts with the in-depth analysis of TLS and B/Plasma cells presented in this manuscript. Maybe "Plasma cells aggregate" or equivalent would be more appropriate.

2) The authors conclude their results section with the following statement. "These findings suggest that high PBR LAs are characterized by terminally exhausted T cell states, consistent with dysfunctional anti-tumor immunity." This is an over-statement as the sole enrichment of exhausted T cells in these discrete niches doesn't prove that overall anti-tumoral immunity is dysfunctional. Furthermore, generation of an in-situ humoral response could likely be anti-tumoral, which is actually stated by the authors themselves in the paragraph starting at line 171. Impact of these structures has not been

determined in the present manuscript.

3) Could the author identify broader impact of the presence of TLS / LAs on the TME? For instance, do high PBR LAs dampen T cells response in the tumor bed (opposed to in the LAs themselves). Do tumor cells undergo higher stress or cell death etc...

4) To further evaluate the role of these high PBR LAs, the authors could look at the isotypes of PCs in the aggregates. Indeed, a majority of IgG PC could indicate anti-tumoral role, while IgA PCs could show pro-tumoral role.

5) Optionally, the authors could perform IgG staining to evidence tumor labelling and indicating likely anti-tumoral role of these high PBR LAs.

6) Could the author find any association of the presence of these structures (whether mature TLS or high PBR LAs) with clinical outcome?

7) Could the authors identify from the pre-treatment samples factors that promoted formation of TLS or high PBR LAs?

Version 1:

Reviewer comments:

Reviewer #4

(Remarks to the Author)

The authors have addressed my comments and significantly improved the clarity of the manuscript. The revised title now more accurately reflects the content. I have no further comments.

REVIEWER COMMENTS

Reviewer #1 (Remarks to the Author):

This is an interesting study looking into the effects of PD-1 blockade with FOLFIRINOX in patients with pancreatic cancer receiving neoadjuvant therapy. The strength of the study is the detailed examination of the resected specimens. The authors have convincingly responded to the reviewers comments. I recommend publication of the article and have no further comments.

We appreciate the Reviewer's unambiguous support.

Reviewer #2 (Remarks to the Author):

This manuscript summarized a pilot study (based on the protocol). The data analysis is mainly descriptive for the primary and secondary outcomes, with several exploratory analysis reported significant associations. Overall, the report is generally well written with several minor issue to be addressed:

The title should state as a pilot trial rather than " A Phase 1 Trial".

We have modified the title to: "FOLFIRINOX Plus Nivolumab Promotes Plasma Cell Production from Intra-Tumoral Lymphoid Aggregates in Borderline Resectable Pancreatic Adenocarcinoma: A Pilot Phase 1 Trial"

The protocol stated that the trial "will be conducted in two parts: where in Part I (6-12 patient safety lead in period). Please briefly describe the results from the part I results in the method.

We have added the following text to the Methods: "After the first 6 patients were enrolled, the safety of the patients was reviewed by the DSMB before proceeding to the second part of the study."

Also, per protocol, "Two interim safety assessments will be performed when first 7 and 15 evaluable patients are enrolled in the study", please also briefly describe the results. There is sample size justification in the protocol, please consider to add it in the method.

The Results section now includes the sentence: "As per the protocol, interim safety assessments were performed when the first 7 and 15 evaluable patients were enrolled in the study. This revealed no dose limiting toxicities and the summary of treatment-related adverse events (TRAEs) is given in **Table 2.**"

Reviewer #4 (Remarks to the Author):

This paper presents the results of a phase 1b/2 trial evaluating neoadjuvant modified FOLFIRINOX plus nivolumab in patients with borderline-resectable PDAC. A deep characterization of the immune microenvironment was performed and lymphoid aggregates containing high amount of plasma cells were described in patients receiving FOLFIRINOX plus nivolumab. Overall, the authors generated a valuable dataset using cutting edge methods and derived observations that could inform immunotherapy developments.

1) Concerning the title: The authors use the term “Irregular” [...] Lymphoid Aggregates”. Usage of the vague term “irregular” contrasts with the in-depth analysis of TLS and B/Plasma cells presented in this manuscript. Maybe “Plasma cells aggregate” or equivalent would be more appropriate.

All but one of our authors greatly appreciate the reviewer’s input on a pre-submission internal disagreement over the word “Irregular”. Thus, we have changed our title to: “FOLFIRINOX Plus Nivolumab Promotes Plasma Cell Production from Intra-Tumoral Lymphoid Aggregates in Borderline Resectable Pancreatic Adenocarcinoma: A Pilot Phase 1 Trial”

2) The authors conclude their results section with the following statement. “These findings suggest that high PBR LAs are characterized by terminally exhausted T cell states, consistent with dysfunctional anti-tumor immunity.” This is an over-statement as the sole enrichment of exhausted T cells in these discrete niches doesn’t prove that overall anti-tumoral immunity is dysfunctional. Furthermore, generation of an in-situ humoral response could likely be anti-tumoral, which is actually stated by the authors themselves in the paragraph starting at line 171. Impact of these structures has not been determined in the present manuscript.

We appreciate this insight and critique of our overstatement. Our intent was to highlight previously published connections between terminally exhausted T cell states (like those we identified) and dysfunctional anti-tumor immunity, yet our statement suggested a broader assessment of anti-tumor immunity and was out of place for a Results section. We have modified our final Results sentence to: “These findings suggest that the PBR of intra-tumoral LAs in this study is connected to production of terminally exhausted T cell states.”

3) Could the author identify broader impact of the presence of TLS / LAs on the TME? For instance, do high PBR LAs dampen T cells response in the tumor bed (opposed to in the LAs themselves). Do tumor cells undergo higher stress or cell death etc...

We appreciate this forward-thinking comment; indeed, we hope to better understand the effects of high-PBR LAs on the proximal tumor microenvironment (especially tumor cells themselves). We partly addressed this relationship in our manuscript by showing that the PBR significantly correlates with the intra-LA ratios of T_{TEX} to T_{PEX} densities (Figure 5D); and that the intra-LA $T_{\text{TEX}}:T_{\text{PEX}}$ ratio generally correlates with that ratio outside of LAs (Figure 5F). These results suggested to us that high PBR LAs may influence T cell phenotypes before

they disperse to the tumor microenvironment. In future studies with larger datasets and animal models, we hope to track T cell subpopulations from their LA origins (especially after PD-1 blockade) to their effects on nearby tumor cells.

4) To further evaluate the role of these high PBR LAs, the authors could look at the isotypes of PCs in the aggregates. Indeed, a majority of IgG PC could indicate anti-tumoral role, while IgA PCs could show pro-tumoral role.

We appreciate the insightful comment. Indeed, antibody isotypes can indicate the potential for antibody-mediated anti-tumor effects. Additionally specific cytokines can differentially promote class switching to specific

isotypes (e.g., TGF- β and IgA). Thus, the inflammatory character of LAs/TLSs can influence humoral anti-tumor immunity. For this reason, we incorporated probes for all IGH isotype (and subsotype) genes into our custom Xenium panel. However, we did not find easily interpretable relationships between the PBR and isotypes of individual cells (**Figure R1A**) nor between the PBR and the dominant isotype of each LA (**Figure R1B**). However, we did find that the density of T_{FH} cells in LAs positively associated with plasma cell (but not B cell) class switching to IgG over IgA (**Figure R2**). In our manuscript, we found a limited negative correlation (P=0.062, Supp Figure 4G) between T_{FH} cell density and LA PBR. It's possible

Figure R2: Follicular helper T cell density in LAs positively correlates with IgG+ Plasma Cells. A) B cells (left panel) and plasma cells (right panel) were assigned isotypes and enumerated from within LAs (classified by T_{FH} cell density tertiles). The number of IgG+ vs IgA+ plasma cells was significantly different by two-tailed Fishers exact test between LA regions of low vs mid T_{FH} cell density (P<0.0001) and mid vs high T_{FH} cell density (P<0.0001)

Figure R1: Isotype class switching is not directly related to the PBR. A) B cells (left panel) and plasma cells (right panel) were assigned isotypes and enumerated from within LAs (classified by PBR score tertiles) and in all analyzed area outside LAs. B) LAs were assigned a dominant class-switched isotype based on the number of B cells or PCs expressing IgG or IgA. **Methods:** Custom Xenium probes were designed for IGHM, IGHD, IGHG1, IGHG2, IGHG3, IGHG4, IGHA1, and IGHA2. IGHM and IGHD transcripts were combined as were the subsotype transcripts for IGHG and IGHA. B cells and plasma cells expressing fewer than 3 IGH transcripts were excluded from the analysis. Each remaining B cell and plasma cell was assigned the IGH isotype with the most transcripts.

that as PBR increases, fewer T_{FH} cells lead to more IgA class switching, but that with extremely high PBR (possibly a consequence of anti-PD-1) the effect is lost (loosely consistent with the strange pattern in the right panel of Figure R1A). However, we considered this level of speculation too distracting and ultimately chose not to present these data in our manuscript.

5) Optionally, the authors could perform IgG staining to evidence tumor labelling and indicating likely anti-tumoral role of these high PBR LAs.

Please see response to #4

6) Could the author find any association of the presence of these structures (whether mature TLS or high PBR LAs) with clinical outcome?

We did not find significant relationships between the abundance of plasma cells, intra-tumoral LA density, LA PBRs, or the densities of T cell subpopulations to overall survival, progression-free survival, pathologic response, or treatment-related changes in CA19-9 and lesion size. The small cohort size in this pilot study may preclude observing important relationships with outcome. Alternatively, in our manuscript, we considered that strong anti-PD-1 activity may promote a beneficial increase in intra-tumoral CD8 T cells, but concomitantly, may also perturb development of adaptive immune responses in LAs (of which abundant plasma cells and high PBR LAs are indicators). PBR-related detrimental effects on patient outcome may be overshadowed by reinvigoration of effector T cells and vice versa, thus limiting significant correlations with patient outcomes.

7) Could the authors identify from the pre-treatment samples factors that promoted formation of TLS or high PBR LAs?

Unfortunately, the pre-treatment biopsies from patients in this study were nearly extinguished for our RNA-seq analyses. We approximated the PBR (for the whole sample) using CIBERSORT scores but did not find a significant correlation between PBRs from the 5 pre-treatment and post-treatment matched pairs ($P=0.68$, two-tailed Spearman correlation, data not shown). Although we did not find evidence of PBR as a biomarker in pre-treatment samples, our sample size and limited material did not allow for a robust investigation.